# Neoadjuvant Immunotherapy in Hormone Receptor-Positive Breast Cancer: From Tumor Microenvironment Reprogramming to Combination Therapy Strategies

**DOI:** 10.3390/ijms262311596

**Published:** 2025-11-29

**Authors:** Zimei Tang, Tao Huang, Tinglin Yang

**Affiliations:** Department of Breast and Thyroid Surgery, Union Hospital, Tongji Medical College, Huazhong University of Science and Technology, Wuhan 430022, China; tangzimeiwh@163.com

**Keywords:** breast cancer, tumor microenvironment, hormone receptor-positive, neoadjuvant immunotherapy, novel therapies

## Abstract

Breast cancer remains the most prevalent malignancy among women worldwide, with hormone receptor-positive (HR+) tumors comprising approximately 70% of cases. Traditionally, HR+ breast cancer has been classified as immunologically “cold” due to its low PD-L1 expression, reduced tumor-infiltrating lymphocytes, and low tumor mutational burden, collectively limiting immunotherapy responsiveness. However, emerging evidence indicates significant molecular heterogeneity within HR+ tumors, characterized by specific genetic signatures and features of the tumor microenvironment (TME) that can be therapeutically reprogramed through chemotherapy-induced immunogenic cell death combined with immune checkpoint inhibition. Recent clinical trials demonstrate that biomarker-selected immune-enriched HR+ subsets, identified by MammaPrint Ultra-High 2 classification, homologous recombination deficiency, or elevated tumor-infiltrating lymphocytes, achieve notable pathological complete response rates with immune checkpoint inhibitor combinations. This review summarizes the dynamic interactions between genetic determinants and TME plasticity in HR+ breast cancer and critically assesses combination strategies across 31 neoadjuvant trials. We demonstrate that optimal efficacy requires biomarker-guided patient selection integrating genetic and TME features, precise sequencing, and a mechanistic understanding of drug-specific immunomodulatory effects. The integration of platform trial designs (I-SPY2, CheckMate-7FL) with composite biomarker algorithms represents a paradigm shift toward precision neoadjuvant immunotherapy, offering a conceptual framework for transforming outcomes in molecularly defined HR+ breast cancer subsets.

## 1. Introduction

Breast cancer is the most prevalent cancer in women globally [1], with hormone receptor-positive (HR+) breast cancer being the most common molecular subtype, constituting approximately 70% of cases [2]. Breast cancer encompasses distinct molecular subtypes: luminal (hormone-driven), HER2-enriched (growth factor dependent), and triple-negative (lacking three receptors). Although immunotherapy has achieved notable success in the more immunogenic triple-negative breast cancer (TNBC), its application in HR+ breast cancer remains challenging. Historically, HR+ breast cancer has been classified as immunologically ‘cold’ due to low programmed cell death ligand 1 (PD-L1) expression, fewer tumor-infiltrating lymphocytes (TILs), and low median tumor mutational burden (TMB), leading to a limited response to immune checkpoint inhibitors (ICIs) [3,4,5]. Nonetheless, recent evidence indicates this cold phenotype might not be absolute. HR+ tumors show considerable molecular heterogeneity that can be targeted therapeutically by reprogramming the tumor microenvironment (TME). Identifying immune-enriched phenotypes with hidden immune potential could greatly benefit patients, as shown by increased pathological complete response (pCR) rates in the CheckMate-7FL trial [6,7]. The traditional view of HR+ breast cancer as uniformly immunologically cold is being reconsidered, revealing its heterogeneous nature and distinct immune-responsive groups.

Neoadjuvant therapy enables tumor downstaging, facilitates breast-conserving surgery, and enhances patient outcomes. However, current standard neoadjuvant chemotherapy shows limited effectiveness in HR+ breast cancer, highlighting the need for novel treatment [8,9]. Combing neoadjuvant immunotherapy with chemotherapy boosts immune responses and alters the TME [10,11]. This approach establishes lasting tumor-specific immune memory, offering ongoing protection against micro-metastatic disease [12,13]. It also increases treatment efficacy, reduces toxicity, and improves patient tolerance [14].

A detailed review of immune potential in HR+ breast cancer reveals it as a heterogeneous disease with substantial immunotherapy-responsive subsets, not just a ‘cold’ tumor. This review summarizes the TME’s role in immune heterogeneity, how combination therapies reprogram immune responses, and precision biomarker for patient selection. It also covers molecular networks for therapeutic synergy and clinical translation. By integrating mechanistic insights with clinical evidence, this review provides a framework for optimizing neoadjuvant immunotherapy in HR+ breast cancer, advancing precision immune-oncology.

## 2. Tumor Microenvironment Heterogeneity and Plasticity in HR+ Breast Cancer: The Static Foundation

HR+ breast cancer not only exhibits lower TMB, but reduced or absent of human leukocyte antigen (HLA) Class I in about 85% of cases [15,16], impairing T cell recognition [17]. Estrogen signaling further decreases HLA Class I expression and silence interferon pathway genes through epigenetic changes, perpetuating immunosuppression [18,19].This antigen presentation defects in HR+ breast cancer are linked to TMB and HLA levels. Notably, HLA Class I downregulation is reversible; interferon-γ (IFN-γ) and tumor necrosis factor-α (TNF-α) can restore its expression, enhancing CD8^+^ T cell activity [20,21]. This suggests that target HLA could convert HR+ tumors into more responsive to immunotherapy [22,23].

In HR+ breast cancer, HLA class I downregulation leads to low tumor-infiltrating lymphocyte (TIL) density, with about 50% of HR+ tumors exhibiting stromal TILs ≤ 10%, unlike higher TIL levels in TNBC [24]. The LETLOB trial shows HR+ breast cancer’s intrinsic heterogeneity: with 26% displaying high-proliferative, immune-enriched gene signatures and significantly increased TIL infiltration [25]. These immune-enriched subsets have more M1 macrophages and activated CD4^+^ memory T cells, but fewer regulatory T cells and M2 macrophages [25]. Estrogen signaling also upregulates indoleamine 2,3-dioxygenase 1 (IDO1) and arginase-1, creating a metabolically hostile microenvironment that hinders TIL infiltration and T cell function [26,27].

Tertiary lymphoid structures (TLS), the presence of organized immune structures, has emerged as a powerful predictor of immunotherapy response in breast cancer [28] [29]. TLS represent ectopic lymphoid organs featuring organized B cell follicles, T cell zones, and high endothelial venules, facilitating local antigen presentation and anti-tumor responses [30]. TLS can range from early aggregates to mature structures with germinal centers, with mature TLS linked to better outcomes and increased immunotherapy effectiveness [31]. Although HR+ breast cancers usually have lower TIL density and fewer TLS than TNBC, the presence of TLS identifies rare “immune-hot” regions within otherwise immunologically cold microenvironments [32]. TLS density correlates with improved outcomes, even in HR+ tumors with low TIL counts, indicating their role in enhancing the effectiveness of infiltrating lymphocytes [33]. The “Immunoscore”, which combines TLS presence, TIL density, and immune cell distribution, offers better prognostic insights than mere cell counts [34]. TLS can be identified through standard H&E staining and confirmed with CD20, CD3, and CD21 markers [30]. Patients with high-density mature TLS are promising candidates for neoadjuvant immunotherapy in HR+ breast cancer, needing further validation in biomarker-driven clinical trials. Figure 1 illustrates the heterogeneity of the HR+ breast cancer TME, highlighting molecular heterogeneity (HLA-I downregulation, PD-L1 and ER expression), cellular diversity (varied immune populations, TAM polarization, and rare TLS) and spatial diversity (immune cell distribution). These features shape the baseline immunological landscape in HR+ breast cancer for potential therapeutic reprogramming.

In HR+ breast cancer, low TIL density is typical, but PD-L1 expression often indicates an adaptive response to existing immune cells rather than inherent tumor expression [35]. This adaptive PD-L1 is linked to higher TIL density and better recurrence-free survival, marking active immune engagement [36]. The spatial relationship between PD-L1+ cells to PD-1+ TILs predicts ICIs response more effectively than individual biomarkers, as effective PD-L1/PD-1 interactions requiring close proximity in TME. PD-L1 expression at the tumor–immune interface is more predictive of pCR than diffuse PD-L1 positivity, highlighting the importance of spatial context [37]. Meanwhile, PD-L1 expression shows significant association with TLS and organized TIL architecture, indicating that ICIs response relies on pre-existing immune organization rather than simple PD-L1 positivity [38]. This spatial dependency requires a combined assessment of PD-L1 expression and TIL distribution and activation to identify HR+ patients who may benefit from checkpoint inhibition [39].

Tumor-associated macrophages (TAMs) play a key role in the immunosuppressive environment of HR+ breast cancer, with high TAM density strongly predicting poor prognosis and resistance to immunotherapy in studies involving 11,000 patients [40,41]. The traditional M1/M2 polarization fails to fully capture TAM complexity in HR+ tumors, where single-cell analyses show a range of activation states influenced by local estrogen levels. These ‘M2-like’ TAMs have increased arginase activity, IL-10 production, and PD-L1 expression [42,43]. Estrogen receptor (ER) signaling promotes TAM recruitment by increasing colony-stimulating factor 1 (CSF-1) and CC-chemokine ligand 2 (CCL2) production, creating a cycle that maintains immunosuppression even during endocrine therapy [44,45].

Beyond immune cells, cancer-associated fibroblasts (CAFs) and abnormal tumor vasculature can also hinder immunotherapy in HR+ breast cancer [46]. CAFs establish physical and biochemical barriers with dense extracellular matrix and immunosuppressive factors (TGF-β, VEGF, CXCL12), aided by estrogen signaling [47]. Abnormal tumor vasculature leads to hypoxic areas that attract regulatory T cells and M2 macrophages, while blocking effector T cells [48]. Normalizing blood vessels could enhance immunotherapy by improving immune cell movement and reducing hypoxia-driven immunosuppression.

## 3. Dynamic TME Reprogramming: Mechanistic Insights into Immunotherapy Response

### 3.1. Chemotherapy-Induced Immunogenic Transformation

Section 2 established the heterogeneous immunosuppressive landscape of HR+ breast cancer. This section explores how chemotherapy leverages the plasticity of TME for immune reprogramming. Chemotherapy achieves dual goals: tumor reduction through cytotoxic effects and immune modulation by enhancing antigen presentation and altering immune cell composition [49,50]. It involves three key sequential mechanistic: ICD induction (Section 3.1.1), antigen processing and presentation enhancement (Section 3.1.2), and immune cells recruitment and spatial reorganization (Section 3.1.3). Critically, chemotherapy also upregulates compensatory checkpoint pathways, laying the groundwork for combined strategies discussed in Section 3.2 [51].

Figure 2 shows the step-by-step reprogramming of the HR+ breast cancer TME during neoadjuvant chemoimmunotherapy. Initially, the immune-cold phenotype (left) has low HLA-I, few TILs, and M2-polarized TAMs. Treatment (middle) combines chemotherapy-induced ICD and checkpoint blockade, initiating TME transformation. The remodeled immune-inflamed state (right) features restored antigen presentation, increased effector T cells, organized TILs, and immunological memory.

#### 3.1.1. Agent-Specific ICD Cascades in HR+ Breast Cancer

ICD triggers chemotherapy-induced immune activation by releasing damage-associated molecular patterns (DAMPs) that activate antigen-presenting cells and prime adaptive immune responses [52]. Key DAMPs include surface-exposed calreticulin, secreted ATP, and released high mobility group box 1 (HMGB1), which facilitate antigen presentation and immune cell recruitment. Early calreticulin exposure promotes dendritic cell (DC) phagocytosis, while ATP and HMGB1 activate DCs via P2X7 and TLR4 receptors, enhancing co-stimulatory molecules. Activated DCs then secrete chemokines to recruit T cell. Notably, ICD induction capacity shows limited correlation with chemical structure; for instance, while both cisplatin and oxaliplatin form DNA adducts, only oxaliplatin elicits ICD [53,54]. In HR+ breast cancer, three neoadjuvant agents exhibit distinct ICD mechanisms [55].

Anthracyclines, especially doxorubicin, constitute the most potent ICD inducers, promoting HMGB1 release and calreticulin surface exposure [56]. Doxorubicin induces caspase-dependent ICD via topoisomerase II inhibition, with calreticulin/ERp57 complex exposure preceding apoptosis through endoplasmic reticulum stress [57]. DAMP release follows a specific sequence: early calreticulin exposure aids DC recognition, ATP secretion activates P2X7 receptors, and later HMGB1 release enhances antigen presentation via TLR4 signaling [58,59,60]. Additional mediators include transcription factor A mitochondrial and formyl peptide receptor 1, with FPR1-deficient DCs showing impaired responses [61]. However, its clinical use remains limited by cardiotoxicity, especially when combined anti-HER2 therapies.

Taxanes like paclitaxel trigger ICD by stabilizing microtubule and activating TLR4 pathway, leading to CALR surface exposure and NF-κB-mediated CCL2 transcription, which boosts DC maturation and antigen presentation [62]. Combining paclitaxel with celecoxib improves DC maturation by inhibiting prostaglandin E2 biosynthesis, while nanoparticles increase cytokine secretion and T cell infiltration [63,64]. Nanoparticles also improve ICD induction and establish long-term immune memory to prevent recurrence when combined with immunomodulatory agents [65].

Cyclophosphamide enhances immune response by depleting Tregs while preserving effector T cells [66], particularly valuable in HR+ breast cancer due to estrogen-driven Treg accumulation [67]. It also induces ICD via type I interferon, promoting DC proliferation and CD8^+^ T cell cross-presentation [68]. This dual mechanism, combining ICD induction with selective immunosuppressive cell depletion, significantly enhances anti-tumor effects when combined with ICIs [69]. Nanomaterials further amplify their immunogenicity by enhancing bioavailability [70,71].

#### 3.1.2. Enhanced Antigen Processing and Presentation

DAMPs released during ICD initiate cascades that enhance antigen presentation, are cucial since most HR+ cases exhibit reduced HLA class I expression [72]. Chemotherapy-induced DNA damage activates the cGAS-STING pathway, elevating interferon-β (IFN-β) production and upregulating HLA class I components [73]. This response restores antigen presentation in HLA-I-deficient tumors, creating new targets for CD8^+^ T cells and addressing the fundamental presentation deficit characteristic of HR+ breast cancer [74].

Chemotherapy not only restores baseline capacity but also creates new immunogenic epitopes through DNA damage-induced mutations, creating neoantigens and cellular stress responses, leading to neoantigens and cryptic epitopes [75]. These neoantigens can bypass central tolerance, expanding the antigen repertoire in HR+ tumors with low mutational burden [76]. The increased cross-presentation ability of ICD-activated DCs is critical for overcoming tumor antigen presentation defects, as chemotherapy-induced tumor cell death provides abundant antigens for processing by professional antigen-presenting cells. bypassing the need for direct tumor HLA class I expression [77,78].

Chemotherapy also modulates tumor-specific CD8^+^ T cells, with drugs like cyclophosphamide boosting their cytotoxicity and reducing PD-1 expression [79]. The restoration and enhancement of antigen presentation supporting the synergy of chemotherapy and ICIs for effective CD8^+^ T cell targeting of cancer cells [80].

#### 3.1.3. Immune Cell Recruitment and TME Priming

The coordinated effects of ICD and enhanced antigen presentation lead to immune cell recruitment and spatial reorganization. This chemotherapy-induced transformation serves as immune priming, not full activation. Chemotherapy disrupts balance by activating DCs through ICD, triggering chemokine production (CXCL9, CXCL10, CCL5) that attracts CD8^+^ T cells to tumors [81]. Simultaneously, the inflammatory TME promotes M1 macrophage polarization, reduces Treg stability, and supports TLS formation for ongoing immune responses [82]. Indicators of successful priming include increased T cell infiltration, better effector-to-regulatory T cell ratios, partial HLA class I restoration, and TLS emergence [83]. However, this transformation also upregulates immune checkpoint molecules like PD-L1, representing adaptive resistance and setting the stage for ICIs therapy [84].

In summary, chemotherapy-induced immunogenic transformation creates conditions for effective immunotherapy in HR+ breast cancer by inducing ICD, enhancing antigen presentation, and reprogramming TME. However, this immune activation simultaneously triggers compensatory checkpoint pathway upregulation, creating a therapeutic window where T cell priming and immune suppression coexist. Strategic checkpoint blockade can leverage this priming to overcome adaptive resistance and sustain anti-tumor responses [85], as discussed in Section 3.2.

### 3.2. Checkpoint Blockade-Mediated Immune Amplification

While chemotherapy-induced immune activation remains constrained by the compensatory upregulation of inhibitory checkpoint pathways that accompany inflammation [86]. The paradoxical effect recruits and activates CD8^+^ T cells while also triggering inhibitory receptors that dampen the immune response [87]. Immune checkpoint blockade actively remodels the TME to enhance and sustain this initial activation [88]. In HR+ breast cancer, this remodeling process faces challenges like estrogen-mediated immunosuppression, lower immune infiltration, and unique checkpoint molecule expression patterns [5,88].

#### 3.2.1. PD-1/PD-L1 Axis Disruption and T Cell Liberation

The PD-1/PD-L1 axis in HR+ breast cancer is influenced by hormonal factors that affect expression patterns and treatment responses [89]. Although ER signaling can promote PD-L1 transcription, HR+ tumors typically exhibit lower baseline PD-L1 than TNBC due to an immunosuppressive microenvironment [35,90]. This baseline difference creates therapeutic opportunities, as chemotherapy-induced immune activation can dramatically upregulate PD-L1 expression in HR+ tumors, establishing optimal conditions for subsequent ICIs therapy [6,7].

PD-1/PD-L1 blockade triggers a complex cascade extending beyond just T cell activation. Restored T cell functions increase IFN-γ production, which upregulates PD-L1 expression via signal transducer and activator of transcription 1 (STAT1), creating a feedback loop that requires ongoing checkpoint blockade for effectiveness [91]. The temporal dynamics reveal distinct phases: rapid T cell activity in the first two weeks, sustainable effector populations forming over 2–8 weeks, and a maintenance phase beyond eight weeks to prevent immune tolerance [92]. This dynamic PD-L1 expression acts as a biomarker, showing an initial spike and gradual decline, forming an inverted U pattern indicative of immune activation [93].

#### 3.2.2. Reversal of Immunosuppressive Networks

Effective checkpoint blockade in HR+ breast cancer requires not only freeing T cells but also dismantling various immunosuppressive cells that keep the immune response weak [94,95]. Regulatory T cells, abundant in HR+ tumors and contributing to immunosuppression through multiple mechanisms, including IL-10 and TGF-β secretion, are reduced and reprogrammed during checkpoint blockade [96]. PD-1 blockade specifically targets and reduces these intratumoral regulatory T cells while maintaining or increasing effector T cells, crucially altering the balance towards a more effective immune response [97].

Myeloid-derived suppressor cells (MDSC), which suppress T cell function through multiple mechanisms including arginase-1 and inducible nitric oxide synthase production, decrease significantly after checkpoint blockade [98]. This reduction occurs through direct effects, like antibody-dependent cellular cytotoxicity, and indirect effects, such as IFN-γ-mediated differentiation into less suppressive myeloid cells [99].

TAMs also shift from an immunosuppressive M2-like state to a pro-inflammatory M1-like state during effective checkpoint blockade [100], driven by IFN-γ from activated T cells. This shift enhances antigen presentation, inflammatory cytokines production, and tumor cell killing [101]. Repolarized macrophages move closer to direct tumor contact zones, improving immune surveillance and tumor elimination [102].

Checkpoint blockade enhances T cell function through reversing exhaustion-associated molecular programs [103], leading to the downregulation of inhibitory receptors like PD-1, T cell immunoglobulin and mucin domain-containing protein 3 (TIM-3), lymphocyte activation gene 3 (LAG-3), and T cell immunoreceptor with immunoglobulin and ITIM domains (TIGIT), and upregulation of effector molecules such as granzyme B, perforin, and IFN-γ [104]. The emergence of T cell factor 1 (TCF1) high memory precursor cells, which have enhanced self-renewal abilities, indicates a favorable prognosis and sustained treatment responses [105].

#### 3.2.3. Mechanisms of Immune Desert-to-Inflamed Conversion

Checkpoint blockade, combined with chemotherapy-induced immune priming (Section 3.1), promotes ongoing immune activation by remodeling TME. Although HR+ tumors are more challenging due to lower mutational burden and estrogen-driven immunosuppression [106], using chemotherapy-induced ICD with checkpoint blockade can effectively convert these tumors in biomarker-selected populations. However, success depends significantly on the tumor’s intrinsic features and baseline immune contexture.

The conversion process involves transforming multiple TME compartments. Vascular remodeling normalizes tumor vessels, enhancing adhesion molecule expression like intercellular adhesion molecule 1 (ICAM-1) and vascular cell adhesion molecule 1 (VCAM-1), facilitating immune cell entry and forming high endothelial venules for immune priming [29,107]. Concurrently, matrix metalloproteinases (MMPs) from activated immune cells break down dense collagen networks, especially in HR+ tumors [108]. The metabolic environment shifts from nutrient-depleted and lactate-rich to one that supports T cell function with reduced lactate, increased glucose, and normalized pH [109]. Chemokine gradients, particularly CXCL9 and CXCL10 driven by IFN-γ from reinvigorated T cells, organize immune cell positioning and facilitate effective immunological synapses [110].

Clinical evidence from the CheckMate 7FL and KEYNOTE-756 trials shows that successful conversion in HR+ breast cancer correlates with specific baseline features: PD-L1 positivity, with pCR rates over 40% using checkpoint blockade versus 20% with chemotherapy alone; lower ER expression, with tumors expressing less than 10% ER benefiting more; and baseline TILs presence [6,7,111]. Dynamic PD-L1 upregulation after chemotherapy indicates successful immune activation. The variability in these remodeling events underscores the need for biomarker-guided strategies to identify HR+ tumors likely to convert successfully and to optimize treatment sequencing [112].

## 4. Precision Biomarker Strategies: Patient Selection and Treatment Monitoring

HR+ breast cancer immunotherapy response varies dramatically, with pCR rates ranging from 10% in unselected populations to 61% in biomarker-enriched cohorts, highlighting precision patient selection and dynamic treatment monitoring strategies [6,7]. This section establishes a practical biomarker framework integrating baseline patient stratification with real-time response assessment to optimize therapeutic outcomes.

### 4.1. Baseline Biomarkers for Patient Selection

#### 4.1.1. PD-L1 Expression: Assay Selection and Clinical Implementation

Various companion diagnostic assays, each with unique antibody clones and scoring methods, were developed for PD-L1 evaluation in checkpoint inhibitor trials. This includes the VENTANA SP142 assay for atezolizumab, the Dako 22C3 pharmDx assay for pembrolizumab, and the Dako 28-8 pharmDx assay for nivolumab. These assays target different patient groups with varied immunotherapy responses [113,114,115]. Comparative studies show significant differences: the SP142 assay has a 35% positivity rate compared to 76% with 22C3, yet SP142-positive patients have higher pCR rates (44.3% vs. 20.2%), indicating SP142’s stricter criteria may better identify immunogenic tumors [116].

The Combined Positive Score (CPS) measures PD-L1-positive tumor and immune cells, divided by total viable tumor cells, then multiplied by 100 [117]. This is crucial in HR+ breast cancer with low immune infiltration. KEYNOTE trials set CPS ≥ 10 as the standard threshold, with CPS ≥ 20 associated with better responses in selected cohorts [118]. PD-L1 expression varies within HR+ tumors due to patchy immune cell distribution and variable ER expression [115]. Multi-site sampling from 3 to 4 spatially separated core biopsies significantly enhances accuracy, reducing false-negative rates from 25% to <10% [119]. Standardized digital image analysis with spatial profiling further improves prediction accuracy compared to traditional methods [120].

#### 4.1.2. Tumor-Infiltrating Lymphocytes and Immune Architecture

Quantitative TIL assessment distinguishes prognostic strata in HR+ breast cancer: 50% of HR+ tumors have low TILs (<10%) with baseline 10–15% pCR rates; 30% have intermediate TILs (10–40%) achieving 25–35% response rates; and 20% have high TILs (≥40%) demonstrating 45–60% pCR rates when combined with checkpoint inhibitor [5,121]. TLS presence is linked to 2.5 times higher pCR rates than TLS-negative tumors [122]. TIL density offers better immunotherapy response prediction than marginal infiltration [123,124]. An immune contexture scoring system, incorporating TIL density, spatial distribution, and TLS maturation, improves predictive accuracy with hazard ratios of 0.3–0.6 [125].

Immune gene enhances tumor classification beyond histological. The 18-gene T cell inflamed signature outperforms individual biomarkers in predicting HR+ breast cancer immunotherapy [126,127]. This method categorizes tumors into three immune subtypes: Immunity_H (signature > 0.5, 15–20% of cases, >40% response to checkpoint blockade), Immunity_M (signature 0.1–0.5, 25–30% of cases, requires combination strategies), and Immunity_L (signature < 0.1, 60–65% of cases, needs immune priming approaches) [128].

#### 4.1.3. HLA Class I Status and Restoration Potential

Analysis reveals distinct HLA-I loss mechanisms: complete loss (~30% of cases) may require alternative treatments; partial downregulation (~40% of cases) could be reversed by chemotherapy; and heterogeneous loss (~30% of cases) suggests for combination therapies [129].

Chemotherapy-induced DNA damage activates the cGAS-STING pathway, leading to type I interferon production and upregulation of HLA class I components. Studies show that about 60% of initially HLA-deficient HR+ tumors significantly recover expression within 2–3 treatment cycles, peaking at weeks 4–6 post-chemotherapy initiation [130]. Monitoring HLA class I provides a quantitative assessment, with the HLA-I recovery index strongly predicting immunotherapy response (OR 3.2, 95% CI 1.8–5.7), making it a critical biomarker for optimizing treatment timing [131].

#### 4.1.4. Integrated Biomarker Algorithms for Patient Stratification

Single biomarker methods have failed to capture the complexity of immune responses in HR+ breast cancer, necessitating the development of multi-gene signatures that encompass T cell activation, interferon signaling, antigen presentation, and immunosuppressive pathways [126,127]. Composite immune scores, developed using machine learning to integrate PD-L1 expression, TIL density, HLA class I status, immune gene signatures, and TMB, achieve superior predictive accuracy (AUC > 0.85) for immunotherapy response [132,133,134]. Multi-institutional validation shows that the Integrated Immune Score categorizes patients into four risk groups with distinct pCR rates, enabling precise treatment selection [32,135,136].

Implementing complex algorithms in clinical practice requires a balance between thoroughness and practicality. The MammaPrint Ultra-High 2 classification achieved a 61% pCR with pembrolizumab plus chemotherapy, compared to 21% with chemotherapy alone [7]. Likewise, CheckMate-7FL’s biomarker-selected cohort utilizing SP142 PD-L1 assessment and high TIL density achieved a 44.3% pCR versus 20.2% in controls [6].

### 4.2. Dynamic Monitoring During Neoadjuvant Treatment

Dynamic monitoring uses circulating and tissue-based biomarkers to track treatment response in real time. Circulating tumor DNA (ctDNA) clearance kinetics offer treatment insights earlier than conventional imaging [137,138]. Rapid ctDNA decline within 2 weeks indicates an optimal response with pCR > 80%, while persistent or rising levels after cycle 2 indicate resistance, necessitating treatment modification [139].

Peripheral blood immune cell profiling complements this by reflecting systemic immune activation. Flow cytometry of T cell activation markers and regulatory T cell frequencies strongly correlates with pathological response [140]. Patients with over a twofold increase in activated CD8^+^ T cells and more than a 50% reduction in regulatory T cells during the first two treatment cycles show significantly higher response rates (65% vs. 25%, *p* < 0.001). An effector-to-regulatory T cell ratio above 5:1 is associated with favorable outcomes [140]. Patients achieving pCR have significantly lower levels of VEGF-A, IFN-γ, TNF-α, and IL-4 compared to non-responders [141,142].

Serial tumor biopsies provide direct evidence of TME transformation. Longitudinal analysis shows that chemotherapy restores HLA class I expression in about 60% of initially deficient tumors within 2–3 cycles, which strongly correlates with a positive immunotherapy response [17]. The HLA restoration index, a fold-change from baseline, serves as a biomarker for timing immunotherapy, with a >2-fold increase indicating readiness for ICIs. Suppression of proliferation (Ki-67 reduction > 50%) within 2–4 weeks and peak immune activation markers (TIL infiltration, HLA-I upregulation) at 4–6 weeks suggest coordinated biological processes [143]. Successful immune transformation is marked by a >4-fold increase in stromal TIL density, a shift to central infiltration, emergence of CD8^+^/CD4^+^ T cell clusters, and organized lymphoid aggregates [112].

Diffusion-weighted MRI with apparent diffusion coefficient (ADC) mapping demonstrates a strong correlation (r = 0.76, *p* < 0.001) between ADC changes and TIL infiltration, with early treatment ADC decreases indicating increased immune cell recruitment [144,145]. A threshold values for immune activation (ΔADC < −15% from baseline) achieve 82% sensitivity and 78% specificity for predicting pathological response [146]. ^18^F-FDG PET-CT metabolic parameters offer complementary insights, with characteristic early metabolic flare (20–40% SUVmax increase) followed by a decline, signaling successful immune activation [147]. A mid-treatment total lesion glycolysis reduction of over 60% correlates with an over 80% probability of pCR [148,149].

### 4.3. Treatment Decision Algorithms

#### 4.3.1. Baseline Risk-Stratified Treatment Selection

Baseline biomarkers allow for evidence-based treatment stratification, as validated in recent Phase III trials. The optimal candidate group, 15–20% of HR+ patients with PD-L1 CPS ≥ 20 or high TILs (≥40%) and mature TLS, demonstrates pCR rates of 60–70% with standard ICIs [150]. The KEYNOTE-756 trial found that MammaPrint Ultra-High 2 classification achieved 61% pCR with pembrolizumab plus chemotherapy, compared to 21% with chemotherapy alone [7]. The standard candidate group, representing 25–30% of patients with PD-L1 CPS ≥ 10 or intermediate TILs (10–40%) or high immune gene signature (>0.5), achieves 40–50% pCR with immunotherapy, as shown in CheckMate-7FL, where biomarker-selected cohorts had 44.3% pCR versus 20.2% in controls [6]. For patients with homologous recombination deficiency (HRD), adding PARP inhibitors offers additional benefits, with I-SPY2’s durvalumab-olaparib-paclitaxel (DOP) arm achieving 28% versus 14% pCR and a 99.6% graduation probability [111].

#### 4.3.2. Response-Adaptive Treatment Modification

Dynamic biomarker monitoring identifies critical intervention windows for treatment optimization. By week 3, ctDNA clearance and immune profiling can pinpoint optimal responders (ctDNA reduction > 2-log, CD8^+^ T cell activation > 2-fold, Treg reduction >50%) who maintain 80% pCR probability with continued current regimens. Suboptimal responders (ctDNA reduction < 1-log or plateau, minimal immune activation) warrant treatment intensification, such as adding PARP inhibitors for HRD-positive cases [111], increasing checkpoint inhibitor dosing, or altering chemotherapy. Patients with rising or stable ctDNA and disease progression require immediate treatment changes.

A mid-treatment assessment at weeks 6–9 confirms changes in TME. Patients showing over a 2-fold increase in HLA-I, more than 20% stromal TIL infiltration, and TLS formation benefit from checkpoint inhibition due to sustained treatment efficacy. Conversely, those with low HLA-I expression and unchanged TIL density should complete planned chemotherapy cycles to maximize priming effects, with extended ICIs use if residual disease persists. Platform trial data show this adaptive strategy boosts response rates from 62% to 74% while maintaining acceptable toxicity, emphasizing the clinical value of response-adapted strategies in the neoadjuvant phase [151].

These biomarker strategies support precision patient selection, with detailed assessment methods and decision thresholds in Table 1. Section 5 examines how these biomarker-driven approaches translate into combination therapy strategies, with Phase III evidence validating their clinical impact.

## 5. Combination Therapy Strategies: From Clinical Evidence to Practice

Systematic analysis of 31 neoadjuvant immunotherapy trials reveals distinct evidence maturity, from established Phase III methods (Table 2) to promising Phase I/II results and informative failures (Table 3). For HR+ breast cancer, biomarker-driven patient selection proves essential, contrasting sharply with TNBC’s all-comers success.

### 5.1. Phase III Evidence: Validated Combinations

#### 5.1.1. HR+ Disease: Biomarker-Driven Success

The CheckMate-7FL trial (NCT04109066, N = 510) showed that precise patient selection transforms modest population-level benefits into significant clinical outcomes [6]. Combining nivolumab with anthracycline–cyclophosphamide followed by paclitaxel in ER 1–10% or ER ≥ 1% populations, the trial achieved 24.5% versus 13.8% overall pCR (*p* = 0.0021). Notably, PD-L1-positive patients (VENTANA SP142 ≥ 1% immune cells) achieved 44.3% versus 20.2% pCR, a 24.1% absolute benefit similar to TNBC response rates. This suggests that SP142’s lower sensitivity (~35% positive versus ~76% with 22C3) effectively identifies immune-activated tumors [116], while ER 1–10% subgroups may share basal-like characteristics despite hormone receptor positivity. These findings indicate that about 15–20% of HR+ tumors harbor intrinsic immunogenic potential that can be targeted with ICIs.

The KEYNOTE-756 trial (NCT03725059, N = 1240) confirmed the effectiveness of using biomarkers to guide treatment with pembrolizumab and paclitaxel/carboplatin, followed by anthracycline in high-risk Grade 3 patients [7]. The overall pCR was 24.3% versus 15.6% (*p* = 0.00005), with PD-L1-positive patients (CPS ≥ 10 by 22C3) achieving 29.7% versus 19.6%. The study highlights that biomarker-guided selection is a universal principle, not limited to specific protocols, with PD-L1+ or ER-low patients achieving 40–45% pCR. While only 15–20% of HR+ patients benefit significantly, this is impactful given that HR+ breast cancer comprises 70% of cases, making biomarker-guided selection a high-impact strategy.

#### 5.1.2. TNBC Standard as Reference

KEYNOTE-522 confirmed pembrolizumab-chemotherapy as the TNBC standard, achieving a 64.8% pCR compared to 51.2% in unselected patients [118], supported by IMpassion031 results (atezolizumab, 57.6% versus 41.1% pCR) [152]. This success is due to TNBC’s inherent immunogenicity, high TMB, elevated TILs, and frequent PD-L1 expression, unlike HR+ which needs biomarker selection to identify immune-responsive subsets in an otherwise immune-cold population.

#### 5.1.3. Clinical Implementation Framework

Phase III trials used two effective strategies: CheckMate-7FL’s anthracycline-first and KEYNOTE-756’s taxane-platinum-first. Both involve checkpoint blockade during chemotherapy-induced immune activation, with the choice guided by patient factors like cardiac risk and immune status. No successful sequential data exists for completing chemotherapy before immunotherapy in Phase III trials. Carboplatin enhances DNA damage-induced immune activation [153]. Clinical application uses biomarker-driven algorithms, with ongoing trials (SWOG S2206, NCT06058377) validating MammaPrint MP2/High-2 selection in Phase III designs.

### 5.2. Phase II Signals: Promising Approaches

#### 5.2.1. PARP Inhibitor Combinations

The I-SPY2 DOP arm achieved 28% versus 14% pCR, with a 99.6% probability of Phase III success [111], leading to SWOG S2206 validation. Critically, over 80% of patients lacked germline BRCA mutations, indicating PARP inhibitors’ effects extend beyond classical synthetic lethality. PARP inhibition enhances immunotherapy via the cGAS-STING pathway activation, increased neoantigens, and restored immune response in HR+ tumors, independent of BRCA mechanisms [154,155]. This is crucial for HR+ breast cancer, where ER signaling suppresses interferon responses.

Patient selection integrates HRD status, BRCA mutations, and immune signatures. HRD-positive patients show increased sensitivity to PARP inhibitors and immune activation. Further trials (OlympiaN NCT05498155, Chinese NCT05761470) confirm the efficacy of combining PARP inhibitors, immunotherapy, and chemotherapy for biomarker-selected HR+ patients.

#### 5.2.2. Antibody–Drug Conjugate Immunotherapy Combinations

Antibody–drug conjugates deliver targeted payload delivery with bystander-mediated tumor killing, generating widespread ICD that produces abundant neoantigens and DAMPs [156]. This approach is promising for HR+ breast cancer, where low immunogenicity limits checkpoint inhibitor effectiveness. The TROPION-Breast04 trial (NCT06112379) is testing datopotamab deruxtecan with durvalumab against the standard KEYNOTE-522 in TNBC and HR-low/HER2- disease, building on datopotamab’s strong performance and potential for immunotherapy synergy. Additional trials are exploring this strategy across various targets and regions, with early safety data showing manageable toxicity. For HR+ breast cancer, these combinations could boost immune activation and improve outcomes in resistant populations. However, Phase III validation remains pending, with TROPION-Breast04 results anticipated as critical proof-of-concept.

### 5.3. Failed Strategies: Lessons for Future Development

#### 5.3.1. CDK4/6 Inhibitor Combinations

CDK4/6 inhibitors combined with aromatase inhibitors (CDK4/6 inhibitor + AI) demonstrate efficacy in metastatic HR+ breast cancer, with real-world evidence showing improved overall and progression-free survival compared to AI alone [157,158]. However, their combination with ICIs faces fundamental mechanistic challenges in the neoadjuvant setting. The striking absence of successful CDK4/6 inhibitor–immunotherapy combinations across 31 trials represents critical development failure, with only CheckMate 7A8 (nivolumab plus palbociclib plus anastrozole) progressing to Phase Ib/II without showing efficacy. This is surprising given the proven metastatic benefits of CDK4/6 inhibitors and preclinical data suggesting they could activate the immune system [159].

The main issue is mechanistic incompatibility: while CDK4/6 inhibition causes cellular senescence and a secretory phenotype, but also profoundly suppresses T cell proliferation, which is essential for ICIs’ efficacy. Clinical trials show severe lymphopenia and impaired T cell function [160], with increased risks of interstitial lung disease and pneumonitis, including fatal cases [161]. Early combination trials with ICIs reporting prohibitive toxicity, leading to discontinuation, suggest fundamental pharmacologic incompatibility.

These trials provide important insights into combination design principles: preclinical synergy does not guarantee clinical benefit, successful combinations require a mechanistic understanding of immunomodulatory effects beyond non-overlapping toxicities, and sequential rather than concurrent administration may be necessary for CDK4/6 inhibitor and immunotherapy combinations.

#### 5.3.2. Other Instructive Failures

In the I-SPY2 trial, the failure of durvalumab plus chemotherapy (without PARP inhibitor) provided an instructive contrast to the DOP arm’s success. Single-agent checkpoint blockade plus chemotherapy proved insufficiency in immune-cold HR+ tumors, while the triple combination with PARP inhibitor worked. This supports strategies requiring immune priming beyond chemotherapy’s immunogenic effects before checkpoint blockade achieves meaningful benefit. Sequential endocrine-immunotherapy approaches lack definitive data, with theoretical concerns about hormone suppression dampening immune activation during checkpoint blockade. Current evidence does not support these strategies outside trials.

Overall, biomarker-driven selection is essential for HR+ immunotherapy, with validated Phase III combinations, promising PARP inhibitor additions in HRD+ subsets, and new antibody–drug conjugate strategies. Failed CDK4/6 combinations highlight the need for understanding mechanisms. Clinical decisions should prioritize validated combinations for biomarker-positive patients, promising trial additions, and emerging strategies awaiting Phase III validation.

## 6. Clinical Translation and Future Directions

To implement neoadjuvant immunotherapy effectively, we need frameworks for validated combinations, innovative trial designs, and new technologies. Success hinges on developing infrastructure, optimizing costs, and continuously innovating in patient selection and treatment delivery.

### 6.1. Development Innovations

I-SPY2’s Bayesian adaptive design allows for efficient combination screening, with the DOP arm advancing to Phase III with a 99.6% probability after testing only 52 patients, compared to the 200–300 patients in conventional designs. By using biomarker stratification, shared controls, and parallel evaluations, timelines are reduced from 8 to 10 years to 3–4 years [162]. CheckMate-7FL used adaptive biomarker-driven enrollment enrichment to focus on ER-low and PD-L1-positive groups as they showed superior responses [6]. SWOG S2206 successfully moved from platform discovery to Phase III validation.

Failures, like CDK4/6 inhibitor combinations blocking T cell proliferation, highlight the need for mechanistic understanding in combination attempts [163]. Single-agent checkpoint blockade with chemotherapy (I-SPY2 durvalumab) was ineffective, while a PARP inhibitor triple combination succeeded, suggesting that immune-cold HR+ tumors need more than chemotherapy’s immunogenic effects for priming. All successful Phase III combinations employed concurrent checkpoint blockade during chemotherapy (not sequential approaches), proving particularly critical in HR+ breast cancer, where chemotherapy-induced immune activation requires immediate checkpoint inhibitor presence, preventing rapid PD-1/PD-L1-mediated suppression.

### 6.2. Future Directions

Emerging strategies with near-term validation potential focus on novel mechanisms and refined selection. antibody–drug conjugate and immunotherapy combinations are the most advanced, with TROPION-Breast04 testing datopotamab deruxtecan plus durvalumab in Phase III against the KEYNOTE-522 standard. Antibody–drug conjugates target the HR+ immune-cold phenotype through bystander-mediated tumor killing, generating neoantigens and DAMPs, potentially expanding immunotherapy-responsive populations beyond current biomarker-selected subsets. Results anticipated for 2026–2027 will determine if this mechanism can overcome baseline immunosuppression in broader HR+ populations.

Ongoing validation studies refine biomarkers for better selection. SWOG S2206 examines the MammaPrint MP2/High-2 classification to identify immunotherapy candidates beyond PD-L1 criteria. ctDNA monitoring is moving from exploratory to validated use, with standardized tests and treatment algorithms allowing real-time response assessment and early intervention for non-responders. Combining tissue-based molecular signatures with liquid biopsy dynamics will lead to composite algorithms that integrate multiple predictive features, enhancing current PD-L1/ER frameworks to improve response rates and reduce unnecessary toxicity.

## 7. Conclusions

With an enhanced comprehension of TME heterogeneity, immunotherapy for HR+ breast cancer has transitioned from empirical investigation to evidence-based precision medicine. Phase III clinical trials have demonstrated that biomarker-selected populations, identified through PD-L1 expression, low ER status, or specific gene expression signatures, achieve pCR rates of 40–45%. This responsive subset constitutes 15–20% of all HR+ breast cancer patients, representing a significant number of individuals due to the high prevalence of breast cancer. In addition to established ICIs combined with chemotherapy combinations, emerging evidence suggests that the inclusion of PARP inhibitor in HRD-positive subsets and the use of antibody–drug conjugate plus immunotherapy approaches may potentially broaden the responsive patient population. Platform trial designs have demonstrated efficient pathways from conceptualization to Phase III validation, thereby accelerating development timelines. Conversely, failed strategies, particularly CDK4/6 inhibitor combinations, underscore that mechanistic understanding must guide rather than follow clinical development.

The integration of advanced biomarker algorithms, validated combination strategies, and adaptive trial methods forms a foundation for ongoing progress in HR+ neoadjuvant immunotherapy. Future developments hinge on validating composite biomarkers, testing new mechanisms, and applying platform trial innovations in practice. Overall, immunotherapy holds the promise of enhancing the efficacy of neoadjuvant therapy in HR+ breast cancer.

## Figures and Tables

**Figure 1 ijms-26-11596-f001:**
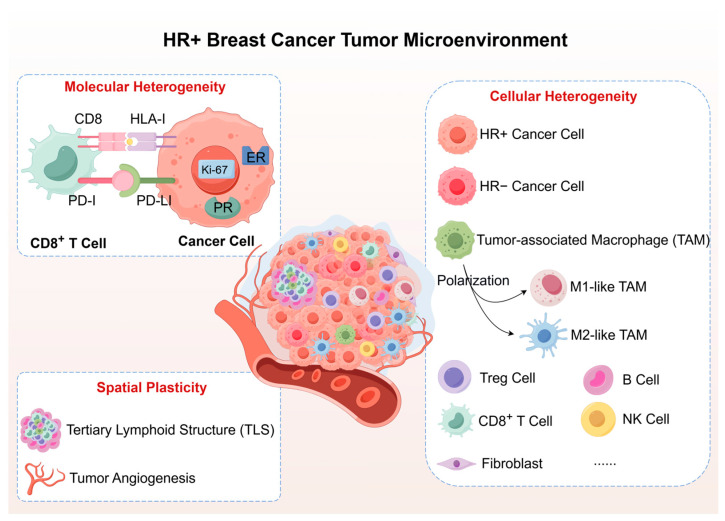
Multidimensional Heterogeneity of the Tumor Microenvironment in Hormone Receptor-Positive Breast Cancer. Hormone receptor-positive (HR+) breast cancer exhibits substantial heterogeneity across molecular, cellular, and spatial dimensions. Molecular features include variable human leukocyte antigen class I (HLA-I) expression, programmed death-ligand 1 (PD-L1) levels, and estrogen receptor (ER)/progesterone receptor (PR) status. Cellular composition features sparse tumor-infiltrating lymphocytes (TILs, including regulatory T cells (Tregs), B cells, CD8^+^ T cells, and NK cells) and tumor-associated macrophages (TAMs) with heterogeneous M1/M2 polarization states, and rare tertiary lymphoid structures (TLS). This baseline heterogeneity determines immunotherapy responsiveness and reprogramming potential. Abbreviations: ER, estrogen receptor; HLA-I, human leukocyte antigen class I; HR+, hormone receptor-positive; PD-L1, programmed death-ligand 1; PR, progesterone receptor; TAM, tumor-associated macrophage; TLS, tertiary lymphoid structure; Treg, regulatory T cell.

**Figure 2 ijms-26-11596-f002:**
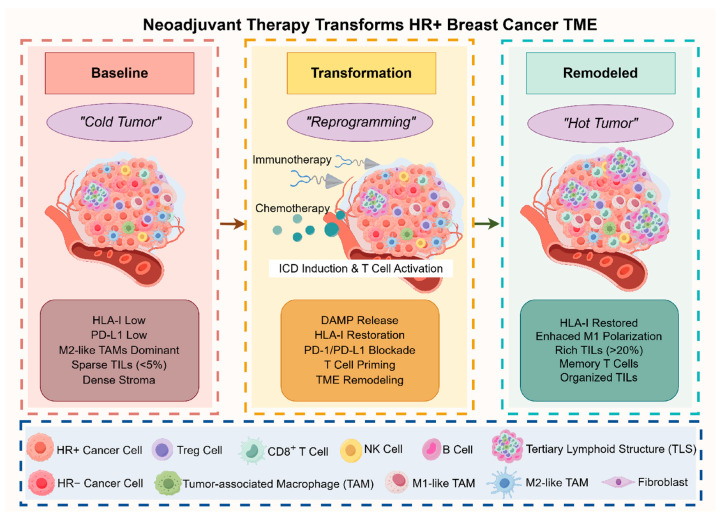
Sequential Reprogramming of the Tumor Microenvironment During Neoadjuvant Chemoimmunotherapy in HR+ Breast Cancer. Neoadjuvant chemoimmunotherapy transforms the tumor microenvironment (TME) from an immune-cold to an immune-inflamed phenotype. The baseline state (left panel) exhibits low human leukocyte antigen class I (HLA-I) expression, sparse tumor-infiltrating lymphocytes (TILs), and M2-polarized tumor-associated macrophages (TAMs). Therapeutic intervention (middle panel) combines chemotherapy-induced immunogenic cell death (ICD) with immune checkpoint blockade targeting programmed cell death protein 1/programmed death-ligand 1 (PD-1/PD-L1), initiating TME transformation. The remodeled state (right panel) features restored HLA-I expression, enriched CD8^+^ T cells, M1-polarized TAMs, and tertiary lymphoid structures (TLS), enabling pathological complete response in biomarker-selected patients. Abbreviations: HLA-I, human leukocyte antigen class I; HR+, hormone receptor-positive; ICD, immunogenic cell death; PD-1, programmed cell death protein 1; PD-L1, programmed death-ligand 1; TAM, tumor-associated macrophage; TIL, tumor-infiltrating lymphocyte; TLS, tertiary lymphoid structure; TME, tumor microenvironment.

**Table 1 ijms-26-11596-t001:** Biomarkers for Neoadjuvant Immunotherapy Patient Selection and Treatment Monitoring in HR+ Breast Cancer *.

Biomarker	Method	Decision Threshold	Supporting Evidence	Reference Section
**Baseline Tumor Selection**				
**PD-L1 expression**	IHC (SP142 or 22C3)	Positive (CPS ≥ 10–20)	SP142 ≥ 1%: 44.3% pCR vs. 20.2% (CheckMate-7FL); 22C3 CPS ≥ 10: 29.7% pCR vs. 19.6% (KEYNOTE-756)	Section 4.1.1
**Tumor-infiltrating lymphocytes (TILs)**	H&E stromal assessment	High density (≥40%)	≥40%: 45–60% pCR (I-SPY2, CheckMate-7FL); 10–40%: 25–35% pCR; <10%: 10–15% pCR (KEYNOTE-756)	Section 4.1.2
**Tertiary lymphoid structures (TLS)**	H&E + IHC (CD20/CD3/CD21)	Presence of mature TLS	TLS-positive: 2.5-fold higher pCR rates (IMpassion031, meta-analysis)	Section 4.1.2
**HLA class I expression**	IHC	Preserved or restorable	Baseline ≥ 50%; Restoration > 2-fold within 2–3 cycles predicts response (I-SPY2)	Section 4.1.3
**ER expression level**	IHC	Low (1–10%)	ER 1–10%: 24.5% pCR vs. 13.8% (CheckMate-7FL); ER ≤ 10% + PD-L1+: 44% pCR	Section 5.1.1
**Genomic signatures**	MammaPrint, gene expression	MP Ultra-High 2	MP2/High-2: 61% pCR vs. 21% control (I-SPY2 pembrolizumab); graduated with 99.6% probability	Section 4.1.4
**Favorable TME Cellular Composition**				
**M1 macrophages**	IHC (CD68+iNOS+CD86+)	High M1 density	High M1/M2 ratio (>1.5) associated with improved response (multiple cohorts)	Section 2
**M2 macrophages**	IHC (CD163+CD206+)	Low M2 density	Low M2/M1 ratio (<0.7); M2 > 40% TAMs predict poor response (meta-analysis, n = 12,439)	Section 2
**Regulatory T cells (Tregs)**	IHC (FOXP3+)	Low Treg infiltration	Treg/CD8^+^ < 0.2; Effector/regulatory > 5:1 predicts 65% vs. 25% pCR (KEYNOTE-522)	Section 3.2.2
**Myeloid-derived suppressor cells (MDSC)**	Flow cytometry (blood)	Low circulating MDSC	MDSC < 5% of PBMCs indicates reduced immunosuppression (I-SPY2)	Section 3.2.2
**CD8^+^ T cells**	IHC (CD8)	High density, central localization	Intratumoral CD8^+^ > 20%: 46% pCR vs. 15%; Central > marginal distribution (TNBC data, applicable to high-TIL HR+)	Section 3.1.2
**Dynamic Monitoring During Treatment**				
**Serial tumor biopsies**	IHC panel (HLA-I, TILs, PD-L1)	Increasing TILs, restored HLA-I, TLS formation	>4-fold TIL increase baseline to cycle 3; HLA-I > 2-fold correlates with pCR (I-SPY2)	Section 4.2
**T cell activation markers**	Flow cytometry (peripheral blood)	Activated CD8^+^ T cells (CD25+Ki-67+)	>2-fold increase CD8+CD25+Ki-67+ within 2 cycles; Effector/Treg > 5:1: 65% vs. 25% pCR (CheckMate-7FL)	Section 4.2
**Circulating tumor DNA (ctDNA)**	Liquid biopsy	Rapid clearance	>2-log reduction by week 3 predicts 80% pCR vs. 20% slow clearance (I-SPY2 ctDNA analysis)	Section 4.2
**PD-L1 dynamic expression**	Serial IHC	Treatment-induced upregulation	On-treatment upregulation indicates immune engagement; inverted U pattern (peak cycle 2–3) optimal (KEYNOTE-522, CheckMate-7FL)	Section 4.2
**Imaging biomarkers**	MRI (ADC), PET-CT (SUVmax)	ADC decrease, metabolic changes	ADC > 15% decrease by week 3: 82% sensitivity; SUVmax > 60% reduction: 75% PPV for pCR (meta-analysis)	Section 4.2

* This table consolidates biomarkers for patient selection and treatment monitoring in neoadjuvant immunotherapy for HR+ breast cancer. Decision thresholds are derived from Phase III trials and I-SPY2 platform analyses current as of October 2025. Abbreviations: ADC, apparent diffusion coefficient; CPS, Combined Positive Score; ctDNA, circulating tumor DNA; ER, estrogen receptor; HLA, human leukocyte antigen; H&E, hematoxylin and eosin; IHC, immunohistochemistry; iNOS, inducible nitric oxide synthase; MDSC, myeloid-derived suppressor cells; MP2/High-2, MammaPrint Ultra-High 2; MRI, magnetic resonance imaging; PBMC, peripheral blood mononuclear cell; pCR, pathological complete response; PD-L1, programmed death-ligand 1; PET-CT, positron emission tomography-computed tomography; PPV, positive predictive value; SUVmax, maximum standardized uptake value; TAM, tumor-associated macrophage; TIL, tumor-infiltrating lymphocyte; TLS, tertiary lymphoid structure; TME, tumor microenvironment; TNBC, triple-negative breast cancer; Treg, regulatory T cell.

**Table 2 ijms-26-11596-t002:** Phase III Neoadjuvant Immunotherapy Trials in Breast Cancer *.

Type of BC	Stage	Trial/NCT Number	Treatment Arms	N	Biomarker	Primary Endpoint	Results
**HR+/HER2− (T1c-2 N1-2 or T3-4 N0-2;** **ER** **≥** **1%)**	Stage II–III	CheckMate-7FL (NCT04109066)	A: Nivolumab + T-AC → Surgery → Nivolumab + ET	263	ER 1–10% or ER ≥ 1%	pCR Rate	Completed—pCR: 24.5% vs. 13.8% (*p* = 0.0021); higher benefit in PD-L1+ subgroup (VENTANA SP142 ≥ 1%: 44.3% versus 20.2%, respectively)
			B: Placebo + T-AC → Surgery → Placebo + ET	258			
**HR+/HER2− (grade 3 high-risk invasive breast cancer (T1c-2, cN1-2 or T3-4, cN0-2))**	Stage II–III	KEYNOTE-756 (NCT03725059)	A: Pembrolizumab + Paclitaxel/Carboplatin → AC → Surgery → Pembrolizumab	635	Grade 3, high-risk	pCR (ypT0/Tis ypN0) + EFS	Ongoing—pCR: 24.3% vs. 15.6% (*p* = 0.00005); higher benefit in PD-L1+ (29.7% vs. 19.6%); EFS was not mature in this analysis.
			B: Placebo + Paclitaxel/Carboplatin → AC → Surgery → Placebo	643			
**HR+/HER2−**	Stage II–III	SWOG S2206 (NCT06058377)	A: Durvalumab + neoadjuvant AC + paclitaxel followed by adjuvant ET	N/A	MP2/High-2	EFS	Recruiting—Testing durvalumab in high-risk HR+ BC based on I-SPY2 results
			B: Placebo + neoadjuvant AC + paclitaxel followed by adjuvant ET	N/A			
**TNBC**	Stage II–III	KEYNOTE-522 (NCT03036488)	A: Pembrolizumab + Paclitaxel/Carboplatin → AC → Surgery → Pembrolizumab	784	PD-L1 (all patients included regardless of status)	pCR + EFS (dual primary)	Completed—FDA approved standard of care; pCR: 64.8% vs. 51.2%; OS HR = 0.66
			B: Placebo + Paclitaxel/Carboplatin → AC → Surgery → Placebo	390			EFS HR = 0.63; 5-year OS: 86.6% vs. 81.7%
**TNBC**	Early Stage	IMpassion031 (NCT03197935)	A: Atezolizumab + Nab-paclitaxel → AC	165	ctDNA	pCR rate	Completed-pCR: 57.6% vs. 41.1%; improved pCR in PD-L1+ patients (69% vs. 49%)
			B: Placebo + Nab-paclitaxel → AC	168			
**TNBC/HR-low/HER2−**	Stage II–III treatment-naive	TROPION-Breast04 (NCT06112379)	A: Datopotamab deruxtecan + Durvalumab → Surgery → Durvalumab	N/A	N/A	pCR, EFS	Ongoing-Challenge to KEYNOTE-522 standard
			B: Standard of care (Pembrolizumab + chemotherapy per KEYNOTE-522)	N/A			Based on BEGONIA study (ORR 79%)
**TNBC**	Early Stage	NCT06627712	A: SBRT + PD-1 inhibitor + Chemotherapy	Recruiting	N/A	pCR rate, Safety	Not yet recruiting—Novel radiotherapy + immunotherapy combination

* This table summarizes completed and ongoing Phase III randomized controlled trials evaluating immune checkpoint inhibitors in the neoadjuvant setting for breast cancer. Data are current as of October 2025. Abbreviations: AC, anthracycline-cyclophosphamide; ctDNA, circulating tumor DNA; EFS, event-free survival; ER, estrogen receptor; ET, endocrine therapy; FDA, Food and Drug Administration; HER2, human epidermal growth factor receptor 2; HR, hazard ratio; HR+, hormone receptor-positive; MP2/High-2, MammaPrint Ultra-High 2 classification; N/A, not available/not applicable; ORR, objective response rate; OS, overall survival; pCR, pathological complete response (ypT0/Tis ypN0); PD-1, programmed cell death protein 1; PD-L1, programmed death-ligand 1; SBRT, stereotactic body radiation therapy; T-AC, paclitaxel followed by anthracycline-cyclophosphamide; TNBC, triple-negative breast cancer.

**Table 3 ijms-26-11596-t003:** Phase I/II Neoadjuvant Immunotherapy Trials in Breast Cancer *.

Type of BC	Stage	Trial/NCT Number	Treatment Arms	N	Biomarker	Primary Endpoint	Results
**PHASE II**							
**HR+/HER2−**	Stage II–III	I-SPY2 (NCT01042379)	A: Pembrolizumab + Neoadjuvant chemotherapy	40	13 mRNA markers	pCR rate	pCR: 30% vs. 13% (control)
			B: Neoadjuvant chemotherapy alone (control)	96			
**HR+/HER2−**	Stage II–III	I-SPY2 (NCT01042379)	A: Durvalumab + Olaparib + Paclitaxel → AC	52	13 mRNA markers	pCR rate	pCR: 28% vs. 14% (control)
			B: Control (NACT): paclitaxel → AC	299			
**HR+/HER2−**	Stage II–III	I-SPY2 (NCT01042379)	Durvalumab + T-AC	N/A	MammaPrint high-risk	pCR rate	Did not graduate—insufficient efficacy signal
**HR+/HER2−** **(Luminal B-like only)**	Early Stage	GIADA	Sequential neoadjuvant chemotherapy followed by nivolumab + endocrine therapy	N/A	Luminal B	Safety, Feasibility	Completed—acceptable safety profile
**HR+/HER2−**	Stage II–III	NCT06639672	PD-1 inhibitor + Chemotherapy + Different RT fractionations (4 arms)	Recruiting	N/A	pCR rate	Not yet recruiting—Immunotherapy + RT fractionation study
**HR+/HER2-low and HER2−**	T1b-c N0 or T1 N1	OlympiaN (NCT05498155)	A: Olaparib + Durvalumab	N/A	gBRCA mutation	pCR rate	Recruiting—PARP inhibitor + immunotherapy
			B: Olaparib alone	N/A			
**HR-low/HER2−**	Early Stage	NCT05749575	Chidamide + Toripalimab + Paclitaxel	28	Low HR expression	tpCR rate (ypT0/is, ypN0)	Unknown status
**HER2−**	Stage II–III	NCT05761470	Camrelizumab + Fluzoparib + Nab-paclitaxel	N/A	HRD	pCR rate	Chinese trial—PARP + PD-1 combination
**HER2-low**	Stage II–III	NCT05726175	Disitamab vedotin (RC48) + Penpulimab (PD-1)	N/A	HER2-low (IHC 1+ or 2+/ISH-)	pCR rate	West China Hospital—completed; manageable safety demonstrated
**HER2+ and HER2-low**	Stage III Inflammatory breast cancer	TRUDI (NCT05795101)	Trastuzumab deruxtecan + Durvalumab	Recruiting	HER2+ or HER2-low (IHC 1+/2+)	pCR rate	Dana-Farber/MD Anderson; First antibody–drug conjugate + immunotherapy for IBC
**HER2+**	Early Stage	Keyriched-1 (NCT03988036)	Pembrolizumab + Trastuzumab + Pertuzumab	N/A	HER2+	pCR rate	Ongoing—Immunotherapy + dual HER2 blockade
**TNBC**	High-risk early stage	I-SPY2.2 (NCT01042379)	Datopotamab deruxtecan + Durvalumab	Recruiting	Adaptive biomarker-driven	pCR rate	New I-SPY platform arm; Based on BEGONIA results
**TNBC**	Early Stage	Neo-CheckRay (NCT03875573)	Durvalumab + oledumab + AC + paclitaxel followed by preoperative radiation	N/A	cT1-3 cN-1, ER+ ≤ 5% or grade 3, or MP high risk	Safety run-in, tumor response, pCR, and RCB	Recruiting
**TNBC**	Early Stage	NCT04418154	Toripalimab + Dose-dense EC → Nab-paclitaxel	N/A	N/A	pCR rate	Ongoing—Chinese PD-1 inhibitor with dose-dense chemotherapy
**TNBC (High TILs)**	Early Stage	NCT05556200	Camrelizumab + Apatinib	N/A	High TILs (≥20%)	pCR rate	Ongoing—Chinese PD-1 + anti-angiogenic combination
**TNBC**	Early Stage	NCT06802757	A: Posaconazole + Pembrolizumab + Chemotherapy	Recruiting	N/A	pCR rate	Not yet recruiting—Novel antifungal + immunotherapy combination
**TNBC**	Early Stage	NCT05582499	Camrelizumab + Chemotherapy	N/A	N/A	pCR rate	Ongoing—Chinese Camrelizumab with standard chemotherapy
**TNBC**	Early Stage	NCT07011823	Pembrolizumab + Partial breast irradiation	Recruiting	N/A	Safety, pCR rate	Not yet recruiting—Immunotherapy + partial breast RT
**PHASE I/II**							
**ER+/HER2−**	Early stage	CheckMate 7A8	Neoadjuvant Nivolumab + Palbociclib + Anastrozole	N/A	N/A	Safety, pCR rate	Ongoing—CDK4/6 inhibitor + PD-1 + AI combination
**Luminal B HER2-/TNBC**	Localized	B-IMMUNE (NCT03356860)	Durvalumab + Neoadjuvant chemotherapy	N/A	Luminal B or TNBC	Safety, pCR rate	Completed—18.8% pCR in luminal B, 45.5% in TNBC
**PHASE I**							
**TNBC**	Early Stage	NCT07178171	QL1706 (PD-1/CTLA-4 bispecific) + Short-cycle anthracyclines/taxanes	30	N/A	pCR rate	Not yet recruiting
**TNBC**	Early Stage	NCT03197389	Pembrolizumab (biomarker study)	N/A	Multiple biomarkers	Safety, Biomarker analysis	Completed—Biomarker identification study
**Other-Observational**							
**TNBC**	Early Stage	NCT06448169	Observational study on PD-1 inhibitor sensitivity	200	N/A	pCR rate	Not yet recruiting—Predictive biomarker identification

* This table summarizes exploratory and early-phase clinical trials evaluating novel immune checkpoint inhibitor combinations in the neoadjuvant setting. These trials investigate various combination strategies including PARP inhibitors, antibody–drug conjugates, CDK4/6 inhibitors, anti-angiogenic agents, and radiation therapy. Data are current as of October 2025. Abbreviations: AC, anthracycline-cyclophosphamide; AI, aromatase inhibitor; BC, breast cancer; CDK4/6, cyclin-dependent kinase 4/6; CTLA-4, cytotoxic T-lymphocyte-associated protein 4; EC, epirubicin-cyclophosphamide; ER, estrogen receptor; gBRCA, germline BRCA mutation; HER2, human epidermal growth factor receptor 2; HR+, hormone receptor-positive; HRD, homologous recombination deficiency; IBC, inflammatory breast cancer; IHC, immunohistochemistry; ISH, in situ hybridization; MP, MammaPrint; mRNA, messenger RNA; N/A, not available/not applicable; NACT, neoadjuvant chemotherapy; PARP, poly (ADP-ribose) polymerase; pCR, pathological complete response; PD-1, programmed cell death protein 1; RCB, residual cancer burden; RT, radiotherapy; T-AC, paclitaxel followed by anthracycline-cyclophosphamide; TIL, tumor-infiltrating lymphocyte; TNBC, triple-negative breast cancer; tpCR, total pathological complete response.

## Data Availability

No new data were created or analyzed in this study. Data sharing is not applicable to this article.

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
