# Peer review of "Neoadjuvant Immunotherapy in Hormone Receptor-Positive Breast Cancer: From Tumor Microenvironment Reprogramming to Combination Therapy Strategies"

_ijms, 2025, doi:10.3390/ijms262311596_

Round 1
Reviewer 1 Report
Comments and Suggestions for Authors
A comprehensive, well-written review on HR+ breast cancer tumor heterogeneity and strategies to render them immunogenic and responsive toward neoadjuvant immunotherapy when combined with conventional chemotherapies. Then article nicely summarizes biomarker guided patient selection methods employed in recent clinical trials and new therapeutics combinations that are being explored and developed.
Overall, I did not find any major issues on the paper. However, following minor edits are recommended.
- The locations of PD-1 and PDL-1 are erroneous and should be interchanged. PD-1 is expressed by CD8+ T cells.
- Can arrange the abbreviations at the end of the paper in alphabetical order for reader's convenience. The abbreviation 'CRT' (supposedly means calreticulin) is not defined in the text.
- The sentences in lines 422–424 will read better with some context about the assays, as they seem to appear suddenly.
- Have lines to separate the adjacent rows in tables for better distinction between them.
- Line 508-509: pCR negatively correlating with serum IFN-Y and TNF-a is somewhat paradoxical and inconsistent with the overall theme of the paper considering their roles in immune activation. Any comments? If the argument is valid, consider adding a couple sentences in the corresponding section to explain the anomaly.
Author Response
Dear reviewer,
We feel great thanks for your professional comments concerning our manuscript ‘Neoadjuvant Immunotherapy in Hormone Receptor-Positive Breast Cancer: From Tumor Microenvironment Reprogramming to Combination Therapy Strategies’. These comments are all valuable and very helpful for further improving our paper. We read the comments carefully and have made corresponding corrections, which we hope to meet with your approval.
Yours sincerely,
Zimei Tang, Tao Huang, and Tinglin Yang
The main corrections in the paper and the response to your comments are as follows. All modifications in the manuscript have been marked up by using the ‘track changes’ function in MS Word. Please note the line numbers in this response refer to those in the clean version, when the ‘track changes’ function is closed.
Detailed comments and responses are as follows.
A comprehensive, well-written review on HR+ breast cancer tumor heterogeneity and strategies to render them immunogenic and responsive toward neoadjuvant immunotherapy when combined with conventional chemotherapies. Then article nicely summarizes biomarker guided patient selection methods employed in recent clinical trials and new therapeutics combinations that are being explored and developed.
Overall, I did not find any major issues on the paper. However, following minor edits are recommended.
Comment 1: The locations of PD-1 and PDL-1 are erroneous and should be interchanged. PD-1 is expressed by CD8+ T cells.
Response 1: Thank you for your comment. We sincerely apologize for this error, and the locations of PD-1 and PD-L1 in Figure 1 have been corrected.
Comment 2: Can arrange the abbreviations at the end of the paper in alphabetical order for reader's convenience. The abbreviation 'CRT' (supposedly means calreticulin) is not defined in the text.
Response 2: Thank you for this suggestion. We have reorganized all abbreviations at the end of the manuscript in alphabetical order (page 22). Additionally, we have replaced all instances of "CRT" with the full term "calreticulin" on page 5 to avoid ambiguity and verified that all abbreviations used in tables are properly defined in the text.
Comment 3: The sentences in lines 422–424 will read better with some context about the assays, as they seem to appear suddenly.
Response 3: We agree that providing context would improve readability. We have added background information at lines 347-350 (page 9) explaining why multiple detection methods exist and their underlying principles, before describing the specific assays (line 352-354). This should help readers better understand the rationale for different PD-L1 testing approaches.
Revisions in our manuscript are also attached here.
[line 347-354]
Various companion diagnostic assays, each with unique antibody clones and scoring methods, were developed for PD-L1 evaluation in checkpoint inhibitor trials. This includes the VENTANA SP142 assay for atezolizumab, the Dako 22C3 pharmDx assay for pembrolizumab, and the Dako 28-8 pharmDx assay for nivolumab. These assays target different patient groups with varied immunotherapy responses[113-115]. Comparative studies show significant differences: the SP142 assay has a 35% positivity rate compared to 76% with 22C3, yet SP142-positive patients have higher pCR rates (44.3% vs 20.2%), indicating SP142's stricter criteria may better identify immunogenic tumors[116]
Comment 4: Have lines to separate the adjacent rows in tables for better distinction between them.
Response 4: We have added horizontal lines to separate adjacent rows in all tables, which should enhance readability and visual clarity.
Comment 5: Line 508-509: pCR negatively correlating with serum IFN-Y and TNF-a is somewhat paradoxical and inconsistent with the overall theme of the paper considering their roles in immune activation. Any comments? If the argument is valid, consider adding a couple sentences in the corresponding section to explain the anomaly.
Response 5: We sincerely thank you for identifying this important inconsistency. Upon careful re-evaluation of the cited literature, we have recognized that this statement requires critical contextualization that was insufficiently addressed in our original manuscript.
The finding cited from reference 140 (Jabeen et al., Oncoimmunology 2018) originates from a study of neoadjuvant chemotherapy combined with bevacizumab (anti-VEGF monoclonal antibody). Critically, the authors reported that lower cytokine levels correlated with pCR mainly in the bevacizumab arm, indicating this appears to be a bevacizumab specific phenomenon rather than a generalizable finding of neoadjuvant therapy. Moreover, cytokine measurements were performed at end-of-treatment, reflecting treatment response dynamics rather than baseline immune status.
The immunological mechanisms underlying anti-angiogenic therapy differ fundamentally from those of checkpoint inhibitor-based immunotherapy, which forms the central focus of our review:
- Anti-angiogenic therapy induces vascular normalization and potentially reduces systemic cytokine spillover as tumor burden decreases, with lower end-of-treatment cytokines serving as markers of effective tumor debulking.
- Checkpoint inhibitor therapy aims to sustain and amplify immune activation, where robust cytokine production within the tumor microenvironment (though not necessarily systemically) is therapeutically desirable.
Mentioning this bevacizumab-specific observation without an extensive mechanistic explanation creates confusion and detracts from our review's thematic coherence. As the reviewer astutely noted, this finding appears paradoxical within the context of immunotherapy-induced immune activation that we emphasize throughout the manuscript.
Therefore, we have removed this sentence entirely to maintain scientific accuracy and thematic focus. We appreciate the reviewer's careful attention to this detail, which has improved the manuscript's clarity and internal consistency.
Reviewer 2 Report
Comments and Suggestions for Authors
I found this review a very timely and value report. It is especially valuable since there is such controversy surrounding the treatment of HR positive breast cancer due to the significant variability of ER positivity, strength of positivity, and the composition of the tumor microenvironment in these mostly cold tumors. This reviewer liked the tables of clinical summaries of past and ongoing clinical trials which should be very helpful to clinicians diagnosing and treating HR+ breast cancers. This reviewer only has a few comments to suggest to the authors, including:
- Overall the paper is just too long to keep the reader engaged. It seems to be somewhat repetitious so it is suggested that the authors shorten the written parts of the paper by about 20% to remove these repetitious sections.
- It might help to organize the treatment segments of the paper to those therapies that either directly kill tumor (chemo, radiation) and those that enhance the immunogenicity of the tumors (checkpoint inhibitors, vaccines) or inhibit the tumor-elicited immunosuppression. There will be overlap such as with chemotherapy, some of these drugs that are used such as cyclophosphamide, have dual actions, for instance, in killing chemo-sensitive tumor cells and in deleting Treg, or up-regulating the expression of HLA-class I, etc to remove some of the key mechanisms of immunosuppression that lets the tumor evade the host immunity.
- I surmise that this was written as a chapter for a book and in the following lines, you should substitute "chapter" with "section." See lines 178, 179, 417.
- I found the referencing to be excellent and very informative.
- I also think that you could highlight perhaps in a small table the key IHC findings in the TME that help clinicians with their decision to use neoadjuvant chemo along with immunotherapy (tertiary lymphoid structures, type I Macrophages, low numbers of Tregs and MDSC, and type II Macrophages, higher MHC class I expression, number of TILS, serial tumor biospies, high PD-L1 expression, T-cell activation markers, etc.) This would highlight the prognostic biomarkers/cells that could help with the treatment selection and prognosis.
Author Response
Dear reviewer,
We feel great thanks for your professional comments concerning our manuscript ‘Neoadjuvant Immunotherapy in Hormone Receptor-Positive Breast Cancer: From Tumor Microenvironment Reprogramming to Combination Therapy Strategies’. These comments are all valuable and very helpful for further improving our paper. We read the comments carefully and have made corresponding corrections, which we hope to meet with your approval.
Yours sincerely,
Zimei Tang, Tao Huang, and Tinglin Yang
The main corrections in the paper and the response to your comments are as follows. All modifications in the manuscript have been marked up by using the ‘track changes’ function in MS Word. Please note the line numbers in this response refer to those in the clean version, when the ‘track changes’ function is closed.
Detailed comments and responses are as follows.
I found this review a very timely and value report. It is especially valuable since there is such controversy surrounding the treatment of HR positive breast cancer due to the significant variability of ER positivity, strength of positivity, and the composition of the tumor microenvironment in these mostly cold tumors. This reviewer liked the tables of clinical summaries of past and ongoing clinical trials which should be very helpful to clinicians diagnosing and treating HR+ breast cancers. This reviewer only has a few comments to suggest to the authors, including:
Comment 1: Overall the paper is just too long to keep the reader engaged. It seems to be somewhat repetitious so it is suggested that the authors shorten the written parts of the paper by about 20% to remove these repetitious sections.
Response 1: We sincerely thank you for this valuable feedback regarding manuscript length and readability. We have carefully revised and polished the manuscript to address these concerns through systematic elimination of repetitive content and streamlining of mechanistic descriptions.
Through these revisions, we have reduced the main body of the manuscript by approximately 15% (from 6653 to 5696 words). We received comment from another reviewer that requested more comprehensive coverage of the tumor microenvironment, which necessitated adding content in Section 2.1. To accommodate both reviewers' suggestions, we have:
- Removed redundant descriptions across sections, particularly where similar concepts were reiterated
- Consolidated overlapping mechanistic discussions to present information more concisely
- Simplified complex sentence structures and reduced verbose phrasing through professional language editing services
- Reorganized content to improve logical flow and minimize repetition between sections
These revisions have substantially improved the manuscript's conciseness and readability. We hope this balance between brevity and completeness better serves the manuscript's goals.
Comment 2: It might help to organize the treatment segments of the paper to those therapies that either directly kill tumor (chemo, radiation) and those that enhance the immunogenicity of the tumors (checkpoint inhibitors, vaccines) or inhibit the tumor-elicited immunosuppression. There will be overlap such as with chemotherapy, some of these drugs that are used such as cyclophosphamide, have dual actions, for instance, in killing chemo-sensitive tumor cells and in deleting Treg, or up-regulating the expression of HLA-class I, etc to remove some of the key mechanisms of immunosuppression that lets the tumor evade the host immunity.
Response 2: We appreciate your suggestion to organize treatments by their primary mechanisms (direct cytotoxicity vs. immunomodulation). While we considered this alternative framework, we retained the current temporal-mechanistic organization, which mirrors the biological cascade and neoadjuvant treatment timeline.
According to your suggestion, we have adjusted the following sections to enhance mechanist clarity.
- Clarified the dual-mechanism framework in Section 3.1's introduction (line 180-185, Page 4) by explicitly distinguishing " tumor reduction through cytotoxic effects" from " immune modulation by enhancing antigen presentation and altering immune cell composition" and emphasizing their overlapping nature. The latter focuses on the immune modulation of chemotherapy drugs rather than the cytotoxic effects.
- Divide into structured subsections and expound on the three parts of immunomodulatory mechanisms in chronological order: ICD induction (3.1.1), antigen presentation enhancement (3.1.2), and immune cell recruitment (3.1.3)
- Noted agents with robust dual mechanisms (e.g., cyclophosphamide combining ICD induction with Treg depletion) within the relevant subsections
We hope this could address the reviewer's insight while preserving the manuscript's narrative flow.
Comment 3: I surmise that this was written as a chapter for a book and in the following lines, you should substitute "chapter" with "section." See lines 178, 179, 417.
Response 3: We thank the reviewer for catching this remnant from an earlier draft format. We have performed a comprehensive find-and-replace throughout the manuscript, specifically correcting line 155, 156, and 343, and confirming no additional instances remain.
Comment 4&5: I found the referencing to be excellent and very informative. I also think that you could highlight perhaps in a small table the key IHC findings in the TME that help clinicians with their decision to use neoadjuvant chemo along with immunotherapy (tertiary lymphoid structures, type I Macrophages, low numbers of Tregs and MDSC, and type II Macrophages, higher MHC class I expression, number of TILS, serial tumor biospies, high PD-L1 expression, T-cell activation markers, etc.) This would highlight the prognostic biomarkers/cells that could help with the treatment selection and prognosis.
Response 4&5: Thank you for your valuable comments. Following your suggestion, we have created a new Table 1: Clinically Actionable Biomarkers for Neoadjuvant Immunotherapy Patient Selection and Treatment Monitoring in HR+ Breast Cancer (see in section 4, page 11-12). This table consolidates:
- Baseline selection markers: PD-L1, TILs, TLS, HLA class I, ER level, genomic signatures
- Favorable TME features: M1/M2 macrophage ratios, Treg/CD8+ ratios, MDSC levels
- Dynamic monitoring parameters: ctDNA kinetics, T-cell activation markers, serial biopsies, imaging biomarkers
Each biomarker entry includes assessment methodology, favorable profiles, clinical decision thresholds with specific pCR data from Phase III trials, and cross-references to relevant manuscript sections. We think this addition could directly address the reviewer's goal of enhancing clinical utility for treatment decision-making.
Reviewer 3 Report
Comments and Suggestions for Authors
Breast cancer is the most common cancer in women, with hormone receptor positive (HR+) tumors making up about 70% of cases. Traditionally considered immunologically “cold” and resistant to immunotherapy, HR+ breast cancer is now recognized as molecularly diverse, with certain genetic and TME profiles showing potential for immune reprogramming. Recent trials reveal that biomarker-selected HR+ subsets such as those with MammaPrint Ultra-High 2 status, homologous recombination deficiency, or high lymphocyte infiltration can respond well to immune checkpoint inhibitors combined with chemotherapy. This review by Tang & Yang et al., highlights how integrating genetic and TME features, biomarker-guided patient selection, and adaptive trial designs (e.g., I-SPY2, CheckMate-7FL) can advance precision immunotherapy for HR+ breast cancer. This review contributes to a better understanding of TME heterogeneity in HR+ breast cancer by highlighting how differences in immune activity, genetic features, and molecular profiles can influence treatment response. It demonstrates the importance of identifying specific biomarkers, such as PD-L1 expression, ER status, and gene signatures, can help recognize patient subgroups more likely to benefit from immunotherapy. Overall, the review supports a gradual shift toward a more precise, evidence-based approach to HR+ breast cancer treatment that considers the diversity of the TME but there are some minor concerns regarding the present version of the manuscript.
Minor comments:
1) English editing: While the review presents valuable insights and comprehensive analysis, the manuscript would benefit from thorough English language editing. Several sentences are complex or awkwardly structured, which may affect clarity for the readers. I recommend careful proofreading and professional language revision to enhance the overall flow, coherence, and precision of the text.
2) The review appears to lack discussion of key studies involving aromatase inhibitor-based therapies. Including relevant clinical and real-world evidence on aromatase inhibitor combinations particularly in comparison to newer treatment modalities would strengthen the comprehensiveness and balance of the review. The study from Rugo et al., PMID: 36220852 used data from the Flatiron Health Analytic Database to compare outcomes of patients with HR+/HER2− metastatic breast cancer treated with palbociclib plus an aromatase inhibitor versus alone. Among 2,888 patients treated between 2015 and 2020, those receiving the combination therapy showed significantly better results. Median overall survival was longer (49.1 vs. 43.2 months), and progression-free survival improved (19.3 vs. 13.9 months) compared to AI alone. These findings support the effectiveness of first-line palbociclib plus AI therapy in routine clinical practice for HR+/HER2− metastatic breast cancer (Trial NCT05361655). Another study from et al., PMID: 39754979 describes the real-world study compared overall survival among patients with HR+/HER2− metastatic breast cancer receiving first-line palbociclib, ribociclib, or abemaciclib combined with an aromatase inhibitor. Using data from 9,146 patients in the US, the study found no significant OS differences between the three CDK4/6 inhibitors, even after adjusting for baseline characteristics and in sensitivity analyses. These results suggest that all three CDK4/6i plus AI combinations offer similar survival outcomes, allowing treatment decisions to be guided by other factors such as safety, tolerability, and quality of life. I would highly encourage the authors to add these key studies in the review with a section describing the studies with aromatase inhibitor clinical trials as well.
3) The present review manuscript is missing a comprehensive overview of the breast tumor microenvironment. The authors are encouraged to elaborate on the specific cellular components of the TME such as immune cells, fibroblasts, and endothelial cells and their multifaceted roles in promoting or suppressing tumor progression, metastasis, prognosis, and response to therapy as mentioned in PMID: 34439387, PMID: 38533507, PMID: 38734280.
Author Response
Dear reviewer,
We feel great thanks for your professional comments concerning our manuscript ‘Neoadjuvant Immunotherapy in Hormone Receptor-Positive Breast Cancer: From Tumor Microenvironment Reprogramming to Combination Therapy Strategies’. These comments are all valuable and very helpful for further improving our paper. We read the comments carefully and have made corresponding corrections, which we hope to meet with your approval.
Yours sincerely,
Zimei Tang, Tao Huang, and Tinglin Yang
The main corrections in the paper and the response to your comments are as follows. All modifications in the manuscript have been marked up by using the ‘track changes’ function in MS Word. Please note the line numbers in this response refer to those in the clean version, when the ‘track changes’ function is closed.
Detailed comments and responses are as follows.
Breast cancer is the most common cancer in women, with hormone receptor positive (HR+) tumors making up about 70% of cases. Traditionally considered immunologically “cold” and resistant to immunotherapy, HR+ breast cancer is now recognized as molecularly diverse, with certain genetic and TME profiles showing potential for immune reprogramming. Recent trials reveal that biomarker-selected HR+ subsets such as those with MammaPrint Ultra-High 2 status, homologous recombination deficiency, or high lymphocyte infiltration can respond well to immune checkpoint inhibitors combined with chemotherapy. This review by Tang & Yang et al., highlights how integrating genetic and TME features, biomarker-guided patient selection, and adaptive trial designs (e.g., I-SPY2, CheckMate-7FL) can advance precision immunotherapy for HR+ breast cancer. This review contributes to a better understanding of TME heterogeneity in HR+ breast cancer by highlighting how differences in immune activity, genetic features, and molecular profiles can influence treatment response. It demonstrates the importance of identifying specific biomarkers, such as PD-L1 expression, ER status, and gene signatures, can help recognize patient subgroups more likely to benefit from immunotherapy. Overall, the review supports a gradual shift toward a more precise, evidence-based approach to HR+ breast cancer treatment that considers the diversity of the TME but there are some minor concerns regarding the present version of the manuscript.
Minor comments:
Comment 1: English editing: While the review presents valuable insights and comprehensive analysis, the manuscript would benefit from thorough English language editing. Several sentences are complex or awkwardly structured, which may affect clarity for the readers. I recommend careful proofreading and professional language revision to enhance the overall flow, coherence, and precision of the text.
Response 1: We thank the reviewer for this constructive suggestion. We have undertaken comprehensive language refinement by a professional native speaker specialized in biomedical manuscripts. The revised manuscript demonstrates enhanced clarity, improved flow between sections, and refined scientific terminology throughout.
Additionally, in addressing feedback from other reviewers, we have substantially improved the manuscript through approximately 15% length reduction by eliminating redundant content. We believe these revisions have substantially strengthened the manuscript's scientific rigor and clinical utility.
Comment 2: The review appears to lack discussion of key studies involving aromatase inhibitor-based therapies. Including relevant clinical and real-world evidence on aromatase inhibitor combinations particularly in comparison to newer treatment modalities would strengthen the comprehensiveness and balance of the review. The study from Rugo et al., PMID: 36220852 used data from the Flatiron Health Analytic Database to compare outcomes of patients with HR+/HER2− metastatic breast cancer treated with palbociclib plus an aromatase inhibitor versus alone. Among 2,888 patients treated between 2015 and 2020, those receiving the combination therapy showed significantly better results. Median overall survival was longer (49.1 vs. 43.2 months), and progression-free survival improved (19.3 vs. 13.9 months) compared to AI alone. These findings support the effectiveness of first-line palbociclib plus AI therapy in routine clinical practice for HR+/HER2− metastatic breast cancer (Trial NCT05361655). Another study from et al., PMID: 39754979 describes the real-world study compared overall survival among patients with HR+/HER2− metastatic breast cancer receiving first-line palbociclib, ribociclib, or abemaciclib combined with an aromatase inhibitor. Using data from 9,146 patients in the US, the study found no significant OS differences between the three CDK4/6 inhibitors, even after adjusting for baseline characteristics and in sensitivity analyses. These results suggest that all three CDK4/6i plus AI combinations offer similar survival outcomes, allowing treatment decisions to be guided by other factors such as safety, tolerability, and quality of life. I would highly encourage the authors to add these key studies in the review with a section describing the studies with aromatase inhibitor clinical trials as well.
Response 2: Thank you for highlighting these important studies on CDK4/6 inhibitor plus aromatase inhibitor (CDK4/6i + AI) combinations in HR+ breast cancer. These studies indeed provide valuable clinical and real-world evidence demonstrating the efficacy of this combination approach, particularly in metastatic disease.
We acknowledge the established efficacy of CDK4/6i + AI in metastatic HR+/HER2− breast cancer, including the real-world evidence showing improved overall and progression-free survival compared to AI alone. Specifically, we have incorporated the suggested references (PMID: 36220852; PMID: 39754979) in Section 5.3.1 (line 591, reference 157-158). In the same section, we explain why this successful combination encounters fundamental mechanistic challenges when combined with immune checkpoint inhibitors in the neoadjuvant setting, as evidenced by the CheckMate 7A8 trial (nivolumab + palbociclib + anastrozole), which was discontinued due to severe hepatotoxicity.
Our discussion now includes:
- Recognition of CDK4/6i + AI efficacy in metastatic disease (citing both suggested references);
- Explanation of the mechanistic barrier: CDK4/6 inhibitors profoundly suppress T cell proliferation, which is a mechanism directly antagonistic to checkpoint blockade;
- Current clinical investigation: CheckMate 7A8 as the only trial attempting CDK4/6i-immunotherapy combination in neoadjuvant breast cancer, now terminated due to unacceptable toxicity.
We respectfully note that our review focuses on neoadjuvant immunotherapy strategies in early-stage HR+ breast cancer. The Phase II trial examining pembrolizumab plus AI (NCT02648477) enrolled patients with metastatic HR+/HER2− disease and demonstrated that while the combination was well tolerated, clinical activity was comparable to AI alone (PMID: 36077811). Given the limited evidence in the neoadjuvant setting and the need to reduce manuscript length according to other reviewers' feedback, we have integrated this discussion within the existing CDK4/6i section rather than creating a separate AI-focused section.
Thus, our addition contextualizes the important CDK4/6i + AI evidence while maintaining focus on immunotherapy-centric approaches relevant to the neoadjuvant setting. We believe this revision appropriately addresses the reviewer's suggestion while preserving the manuscript's defined scope. Revisions in our manuscript are also attached here.
[line 589-597]
CDK4/6 inhibitors combined with aromatase inhibitors (CDK4/6 inhibitor + AI) demonstrate efficacy in metastatic HR+ breast cancer, with real-world evidence showing improved overall and progression-free survival compared to AI alone [157,158]. However, their combination with ICIs faces fundamental mechanistic challenges in the neoadjuvant setting. The striking absence of successful CDK4/6 inhibitor-immunotherapy combinations across 31 trials represents critical development failure, with only CheckMate 7A8 (nivolumab plus palbociclib plus anastrozole) progressing to Phase Ib/II without showing efficacy. This is surprising given the proven metastatic benefits of CDK4/6 inhibitors and preclinical data suggesting they could activate the immune system[159].
Comment 3:The present review manuscript is missing a comprehensive overview of the breast tumor microenvironment. The authors are encouraged to elaborate on the specific cellular components of the TME such as immune cells, fibroblasts, and endothelial cells and their multifaceted roles in promoting or suppressing tumor progression, metastasis, prognosis, and response to therapy as mentioned in PMID: 34439387, PMID: 38533507, PMID: 38734280.
Response 3: We thank you for this constructive suggestion. We agree that a comprehensive overview of the breast tumor microenvironment is essential for understanding the mechanisms underlying immunotherapy response in HR+ breast cancer.
We acknowledge that while our review extensively addresses immune cell populations and their spatial organization in Section 2.1, we had provided limited discussion of other critical TME components. In response to the reviewer's feedback, we have expanded Section 2.1 (lines 133-139, Page 3) to include:
- Cancer-associated fibroblasts (CAFs): Added discussion of their dual immunosuppressive role through extracellular matrix deposition, creating physical barriers to T cell infiltration, and secretion of immunosuppressive factors (TGF-β, VEGF, CXCL12). We incorporated insights from PMID: 34439387 and 38533507 regarding CAF activation by estrogen signaling in HR+ breast cancer.
- Tumor vasculature and endothelial cells: Added content on aberrant tumor angiogenesis, creating hypoxic, immunosuppressive microenvironments and impaired immune cell trafficking through abnormal vessel architecture, incorporating reference PMID: 38734280.
These additions specifically emphasize how these stromal and vascular components influence immunotherapy response to maintain our review's focus on actionable insights for neoadjuvant immunotherapy strategies rather than comprehensive TME biology. Given space constraints, we have cited key comprehensive reviews (PMID: 34439387, 38533507, 38734280) to guide readers seeking more detailed TME information and their roles in HR+ breast cancer progression. Revisions in our manuscript are also attached here.
[line 133-139]
Beyond immune cells, cancer-associated fibroblasts (CAFs) and abnormal tumor vasculature can also hinder immunotherapy in HR+ breast cancer [46]. CAFs establish physical and biochemical barriers with dense extracellular matrix and immunosuppressive factors (TGF-β, VEGF, CXCL12), aided by estrogen signaling [47]. Abnormal tumor vasculature leads to hypoxic areas that attract regulatory T cells and M2 macrophages, while blocking effector T cells [48]. Normalizing blood vessels could enhance immunotherapy by improving immune cell movement and reducing hypoxia-driven immunosuppression.
Reviewer 4 Report
Comments and Suggestions for Authors
This is a timely, well-written review addressing neoadjuvant Immunotherapy in Hormone Receptor-Positive (HR+) Breast Cancer. HR+ breast cancer typically shows poor response to immunotherapy, and therefore, there is an urgent need to identify approaches to boosting immunotherapy efficacy and to stratify patients.
The authors provide a well-structured overview of immune microenvironment heterogeneity from a unique perspective, categorizing it into molecular heterogeneity (such as HLA class I downregulation and PD-L1 upregulation), cellular heterogeneity (low TIL infiltration and macrophage immunosuppressive reprogramming), and spatial heterogeneity (such as TLS). This summary illustrates how multiple layers of heterogeneity can affect response to immunotherapies. Therefore, taking into accounts different factors are important in predicting treatment outcomes.
The authors next summarize the mechanisms by which neoadjuvant Immunotherapy turns immune “cold” tumor into immune “hot”, and further review biomarkers for selecting patients. Finally, they provide the clinical evidences of using neoadjuvant immunotherapy for HR+ breast cancer and discuss the challenges in combination strategies for HR+ breast cancer.
Together, this review successfully integrates mechanistic insights and clinical findings. The figures are clear, and the language is good. I recommend acceptance after minor polishing.
Author Response
Dear reviewer,
We feel great thanks for your professional comments concerning our manuscript ‘Neoadjuvant Immunotherapy in Hormone Receptor-Positive Breast Cancer: From Tumor Microenvironment Reprogramming to Combination Therapy Strategies’. These comments are all valuable and very helpful for further improving our paper. We read the comments carefully and have made corresponding corrections, which we hope to meet with your approval.
Yours sincerely,
Zimei Tang, Tao Huang, and Tinglin Yang
The main corrections in the paper and the response to your comments are as follows. All modifications in the manuscript have been marked up by using the ‘track changes’ function in MS Word. Please note the line numbers in this response refer to those in the clean version, when the ‘track changes’ function is closed.
Detailed comments and responses are as follows.
Comment: This is a timely, well-written review addressing neoadjuvant Immunotherapy in Hormone Receptor-Positive (HR+) Breast Cancer. HR+ breast cancer typically shows poor response to immunotherapy, and therefore, there is an urgent need to identify approaches to boosting immunotherapy efficacy and to stratify patients.
The authors provide a well-structured overview of immune microenvironment heterogeneity from a unique perspective, categorizing it into molecular heterogeneity (such as HLA class I downregulation and PD-L1 upregulation), cellular heterogeneity (low TIL infiltration and macrophage immunosuppressive reprogramming), and spatial heterogeneity (such as TLS). This summary illustrates how multiple layers of heterogeneity can affect response to immunotherapies. Therefore, taking into accounts different factors are important in predicting treatment outcomes.
The authors next summarize the mechanisms by which neoadjuvant Immunotherapy turns an immune “cold” tumor into an immune “hot” one, and further review biomarkers for selecting patients. Finally, they provide the clinical evidence of using neoadjuvant immunotherapy for HR+ breast cancer and discuss the challenges in combination strategies for HR+ breast cancer.
Together, this review successfully integrates mechanistic insights and clinical findings. The figures are clear, and the language is good. I recommend acceptance after minor polishing.
Response: We sincerely thank you for this encouraging assessment and for recognizing the timeliness and clinical relevance of our review. We are honored that the reviewer found our multidimensional framework for TME heterogeneity to be well-structured and agreed with our integration of mechanistic insights with clinical evidence.
In response to the suggestion for minor polishing, we have undertaken comprehensive language refinement through professional native English editing services specializing in biomedical manuscripts. This editing has enhanced clarity, terminology consistency, and overall readability throughout the manuscript.
We believe these revisions have substantially improved the manuscript's accessibility and clinical utility for the broad readership this review targets.
Round 2
Reviewer 2 Report
Comments and Suggestions for Authors
This reviewer thanks the authors for their revisions which I believe greatly enhances the value of this paper. No further comments or revisions are requested. Congratulations.